# Polymorphisms Analysis of *BMP15*, *GDF9* and *BMPR1B* in Tibetan Cashmere Goat (*Capra hircus*)

**DOI:** 10.3390/genes14051102

**Published:** 2023-05-18

**Authors:** Tianzeng Song, Yacheng Liu, Renqing Cuomu, Yao Tan, Cuoji A. Wang, Ji De, Xiaohan Cao, Xianyin Zeng

**Affiliations:** 1Institute of Animal Science, Tibet Academy of Agricultural and Animal Husbandry Science, Lhasa 850009, China; songtianzeng123@sina.com (T.S.); renqingcuomu666@126.com (R.C.); acuo219@163.com (C.A.W.); deji0891@163.com (J.D.); 2Isotope Research Laboratory, Sichuan Agricultural University, Ya’an 625014, China; liuyachengsicau@126.com (Y.L.); ustanak0217@163.com (Y.T.)

**Keywords:** Tibetan cashmere goat, SNP, *BMP15*, *GDF9*, *BMPR1B*

## Abstract

The Tibetan cashmere goat is a prolific goat breed in China. In sheep breeds, natural mutations have demonstrated that the transforming growth factor beta (TGF-β) super family ligands, such as growth differentiation factor 9 (*GDF9*), bone morphogenetic protein 15 (*BMP15*) and their type I receptor (bone morphogenetic protein receptor (*BMPR1B*), are essential for ovulation and increasing litter size. In this study, 216 female Tibetan cashmere goats were sampled, and candidate genes with fecundity traits were detected via restriction fragment length polymorphism (RFLP) and sequenced. Four polymorphic loci were found in specific amplification fragments of *BMP15* and *GDF9*. Two SNP sites of the *BMP15* gene were discovered, namely G732A and C805G. The G732A mutation did not cause the change in amino acids, and the frequencies of each genotype were 0.695 for the GG type, 0.282 for the GA type and 0.023 for the AA type. The C805G mutation caused amino acids to change from glutamine to glutamate. The genotype frequencies were 0.620 for the CC type, 0.320 for the CG type and 0.320 for the CG type. For the GG type 0.060, the G3 and G4 mutations of the *GDF9* gene were all homozygous mutations. Two known SNP sites, C719T and G1189A, were detected in the Tibetan cashmere goat *GDF9* gene, of which the C719T mutation caused a change of alanine to valine, with a genotype frequency of 0.944 for the CC type and 0.056 for the CT type, whereas no TT type was found. The G1189A mutation caused valine to become isoleucine, and the frequencies of each genotype were 0.579 for the GG type, 0.305 for the GA type and 0.116 for the AA type; G1, B2, B3, B4, FecX^H^, FecX^I^, FecX^L^, G2, G5, G6, G7, G8, FecG^E^, FecTT and FecB mutations were not found in Tibetan cashmere goats. The results of this study provide a data basis for future studies of *BMP15*, *GDF9* and *BMPR1B* gene mutations in goats.

## 1. Introduction

The profit from goat rearing is determined by two main factors, namely, production and prolificacy [1]. Prolificacy is among the most important traits in goats and is regulated by a number of genes. Prolificacy candidate genes include bone morphogenetic protein 15 (*BMP15*), growth differentiation factor 9 (*GDF9*) and bone morphogenetic protein receptor, type IB (*BMPR1B*), all of which have important functions in regulating ovarian functions in animals. These genes, which affect ovulation and litter size, are collectively referred to as fecundity (Fec) genes. Some specific mutations found on these genes have been shown to be associated with different phenotypic effects. Some recent studies on some prolific goat breeds have suggested that higher prolificacy in goats may differ from that of sheep. Some identified point mutations of the *BMPR1B* (FecB) and BMP15 (FecX^H^, FecX^I^, FecX^G^ and FecX^B^) genes are monomorphic in the prolific goat breeds. *BMP15*, also known as growth differentiation factor 9B, is a growth factor secreted by oocytes in the bone morphogenetic protein subfamily in the transforming growth factor *β* superfamily [2,3]. It is located on the X chromosome and has shown a notable role in sheep ovarian folliculogenesis. Mutations in the *BMP15* gene, such as B1, B2 (FecX^G^), B3, B4 (FecX^B^), FecX^I^, FecX^L^, FecX^H^, FecX^R^, FecX^Gr^, FecX^O^ and others, have been identified in sheep. Additionally, FecX^I^, FecX^B^, FecX^L^, FecX^H^, FecX^G^ and FecX^R^ are identified as fecundity-related mutations [4,5,6,7,8]. The mutation of *BMP15* genes associated with the fecundity in goats has been confirmed in Markhoz goats. In a previous study in sheep, the rate of ovulation in *BMP15* mutants was greater in heterozygotes, whereas the homozygous mutants showed a primary ovarian failure, which led to sterility [9]. In contrast, homozygous mutant animals showed higher numbers of kids in Beetal goats [10]. *GDF9* is a growth factor secreted by oocytes that plays an important role in the growth and differentiation of follicles [11]. The expression of *GDF9* was identified as oocyte-specific in ovine and bovine ovaries, which began at the primordial follicle stage. In sheep, the gene mutation of *GDF9* may cause infertility or an increased ovulation rate. Mutations of *GDF9*, including FecGH, FecGE, FecGSI and FecTT, have been identified to be associated with fertility. In developing ovarian follicles, the *GDF9* growth factor is secreted and plays role in growth and differentiation in early ovarian follicles. Animals homozygous for *GDF9* are known to be anovulatory, while animals heterozygous for *GDF9* have a higher than normal ovulation rate [12]. The genetic polymorphisms of the *GDF9* were shown to affect fecundity traits in farm animals, and the heterozygous genotypes caused an increase in the ovulation rate and enhanced the prolificacy in the farm animals, in comparison to wild-type homozygous mothers [13]. There is a high influence of *BMP15* and *GDF9* on fecundity. These genes are produced by the ovary and influence its function, aside from their involvement in increasing the ovulation rate in goats. They also affect follicle growth and development at all stages of folliculogenesis in females [14]. They induce mitosis along with differentiation in the follicular somatic cells during follicular development through the paracrine signaling pathway. They also play key roles in litter size. Numerous mutations in these genes may contribute to high prolificacy. After years of research, more than a dozen mutations in *GDF9*, such as G1, G2, G3, G4, G5, G6, G7, G8, FecG^E^, FecTT and FecG^V^, have been found in sheep [8,15,16,17]. Among them, G8 mutation heterozygosity increases the number of ovulations, whereas homozygous mutations lead to infertility [7,8]. FecX^B^ has been found in goats. It has also been shown that the genetic polymorphisms of the *GDF9* gene in goats were significantly higher than in sheep and also more complicated. Heterozygous ewes with mutant alleles of *GDF9* and *BMP15* have high ovulation rates compared to those animals carrying the normal homozygous wild genotype. In sheep, it has been shown that homozygous mutants of *GDF9* and *BMP15* cause ovarian failure, which leads to sterility [1].

*BMPR1B* is a membrane receptor for bone morphogenetic proteins that are widely present in different tissues of the body [18]. BMP 2, 4, 6, 7 and 15 are expressed in the ovary as candidate ligands of the BMPR1B receptor. *BMPR1B* is a dominant autosomal gene found in chromosome 6 and ensures fecundity and twinning in small ruminants [19]. Given that the tendency to have twins and triplets is inherited and may be common in sheep and goats, the *BMPR1B* gene was thought to be a potential candidate gene for prolificacy in goats because of its critical role in the regulation of terminal folliculogenesis and the control of ovulation rate [20]. Souza et al. [21] studied the *BMPR1B* gene in sheep with two mutation sites, one for FecB mutation and the other for C1113A mutation. Chu et al. [22] studied the polymorphisms of nine exons of the sheep *BMPR1B* gene and found 20 new mutation sites in addition to FecB and C1113A, of which 3 polymorphism sites, G922T, T1043C and G192A, led to the corresponding amino acid changes but did not affect the litter size. FecB mutation in the *BMPR1B* gene has also been found in many goat breeds. A previous study found them to be homozygous non-carriers. FecB polymorphism in Raighar goats showed the monomorphic status of the breed. In Markhoz goats, it was supposed that the FecB allele might not be the only main cause for the prolificacy rate. Some studies have shown that genes of *BMP15*, *GDF9* and *BMPR1B* play the most important roles in the growth and differentiation of early ovarian follicles and have an important influence on the litter size [23,24,25].

Current research on sheep *BMP15*, *GDF9* and *BMPR1B* has achieved great results and showed some mutations that may affect high fertility traits. However, there are relatively few related studies on goats. *BMP15*, *GDF9* and *BMPR1B* with their related genetic markers affecting the litter size in goats are also relatively lagging and these restrict the progress of goat breeding and development.

The Tibetan cashmere goat, also regarded as the Kashmiri goat [26], is an important economic animal in Tibet with its collection of fur, skin, meat, velvet and milk [27]. These small ruminants withstand harsh environmental conditions. They live in arid and semi-arid areas of Tibet, where they are reared for their benefits. They are also known for reproduction traits with high fecundity, big litter size and precocious sexual maturity. They can, therefore, be used as valuable material to explore genetic markers related to economic traits. Genes and genetic variations associated with goat litter size, *GDF9*, *BMP15* and *BMPR1B* have been identified [28]. In this study, Tibetan cashmere goats in the Arena area of northwest Tibet were used, and gene mutations as well as polymorphisms of *BMP15*, *GDF9* and *BMPR1B* genes were detected via RFLP and PCR product sequencing. This provided us with basic information and data for further research and screening of the main genes affecting the fertility of Tibetan cashmere goats. It also provided a data basis for the main genes affecting the litter size in goats and related genetic markers.

This study focuses on the effects of polymorphism and mutation of genes (*BMP15*, *GDF9* and *BMPR1B*) in Tibetan cashmere goats. It is advantageous to identify genes and mutations that can be important for their breeding, better fecundity and normal physiology to improve productivity and reproductive traits.

## 2. Materials and Methods

### 2.1. Experimental Materials

#### 2.1.1. Laboratory Animals

A total of 216 female Tibetan cashmere goats were randomly selected from seven counties (Ritu, Gaize, Gar, Bangor, Genji, Nyima and Tsoqin) in Tibet’s Ali and Nagqu districts. There were single-birth animals with three to five consecutive parities. All the goats were healthy and aged 3–4 years. They were kept in sheltered outdoor paddock and were provided with alfalfa hay and water available ad libitum.

#### 2.1.2. Main Reagents

A blood/tissue/cell genome extraction kit was purchased from Beijing Tiangen Biotechnology, China. Molecular reagents, such as Taq DNA polymerase and dNTP, were purchased from Dalian TAKARA Corporation (Dalian, Liaoning, China) and used according to the manufacturer’s instructions.

### 2.2. Experimental Methods

#### 2.2.1. Sample Collection

Ear tissues from Tibetan cashmere goats (about 1 cm^3^) were collected in vivo and stored in ethanol solution for DNA extraction, and all samples were temporarily stored and transported in an ice box after collection and finally stored in an ultra-low temperature freezer at −80 °C. All experimental procedures involving the animals were reviewed and approved by the Animal experimental committee of Sichuan Agricultural University. The ethical approval code is (SAU2015068, 12 December 2015).

#### 2.2.2. Primer Design and Synthesis

According to the *BMP15* gene sequence of sheep published in GenBank (accession number: EU743938), the primer premier 5.0 software was used to specifically amplify primers of BMP15-1 for the exon II segment, and its amplification region covered the known B2, B3, B4, FecX^H^, FecX^I^ and FecX^L^ alleles in sheep. Specific primers of GDF9-1 and GDF9-2 were designed for exon fractions based on published sheep *GDF9* gene sequences (accession number: EF446168) using primer premier 5.0 software, and their amplification regions covered known G2, G3, G4, G5, G6, G7, G8, FecG^E^ and FecTT alleles in sheep. The *G1* mutation restriction fragment length polymorphism detection primers reported by Hanrahan et al. [29] and the primers G1-RFLP and BMPR1B-RFLP detection primers reported by Davis et al. [30] in 2002 were referred to. The primers were synthesized by Beijing Qingke Xinye Biotechnology Co., Ltd. and Primer information is available in Table 1.

#### 2.2.3. Genomic DNA Extraction

The collected samples were ground into powder using liquid nitrogen and a mortar, and the genomic DNA was extracted according to the instructions of the tissue genome extraction kit (purchased from Beijing Tiangen Biotechnology, Beijing, China).

#### 2.2.4. PCR Amplification and Detection

The extracted genomic DNA was taken as a template along with primers of GDF9-1, GDF9-2, G1-RFLP, BMP15-1 and BMPR1B-RFLP. The total reaction volume was adjusted to 25 μL, see Table 2. Briefly, the reaction parameters were: predenaturation at 94 °C for 3 min; 94 °C denaturation for 30 s; annealing for 40 s, 72 °C extended for 30, 30, 30, 40 and 20 s, 34 cycles; and finally, 72 °C extended for 10 min and later stored at 4 °C.

The PCR amplification product was detected via 2% agarose gel electrophoresis, 1.5 μL of the PCR amplification product was taken and electrophoresis was performed at 120 V for 30 min; the electrophoresis results were analyzed using a gel imaging system.

#### 2.2.5. Digestion Typing and Sequencing

The restriction enzymes were manufactured by Dalian TAKARA Corporation (Dalian, Liaoning, China). The G to A nucleotide change in GDF9 exon 1 disrupts a HhaI restriction enzyme cleavage site (GCGC to GCAC) at nucleotide 260 of the 462 bp PCR fragment produced by primers G9-1734 and 2175. Restriction digestion of the PCR product from wild-type animals with *HhaI* resulted in cleavage of the 462 bp product (at two internal HhaI resulted in cleavage of the 462 bp products (at two internal HhaI sites)) into fragments of 52, 156 and 254 bp. However, DNA fragments containing the A nucleotide yielded only two fragments (52 and 410 bp). Animals heterozygous for the mutation had fragments of all four sizes (52, 156, 254 and 410 bp). The enzymes used were obtained from a previous publication. The products amplified by G1-RFLP and BMPRIB-RFLP were digested with *HhaI* and *Ava* II enzymes separately, and the total volume was adjusted using products 10 μL, 10× Buffer R 1.5 μL, enzyme (10 U/μL) 0.1 μL, ddH2O 3.4 μL, 37 °C overnight, and the digestion product was electrophoretic typing using 3% agarose in TBE buffer at 150 V for 30 min. The amplification products of GDF9-1/GDF9-2 and BMP15-1 were sent to Shanghai Bioengineering Co., Ltd. (Shanghai, China) for sequencing, and the results were compared and analyzed using Geneious 10.0.5 software.

## 3. Analysis of the Results

### 3.1. PCR Product Detection

The genomic DNA amplification of ear tissues and the products were detected by agarose gel electrophoresis. The results showed that the size of the amplified band was consistent with the target band, and the specificity was good.

### 3.2. Digestion Products

The products amplified by G1-RFLP and BMPR1B-RFLP were digested (Figure 1) with *HhaI* and *Ava* II enzymes. The enzyme, as shown in Figure 2, had *HhaI* enzyme cleavage G1-RFLP product typing bands: GG genotypes (wild type) 254 bp, 156 bp and 52 bp, AA genotypes 414 bp and 52 bp, GA genotypes 414 bp, 254 bp, 156 bp and 52 bp; the *Ava* II enzyme cleavage BMPR1B-RFLP product typing bands were AA genotypes (wild type) 191 bp, GG genotypes 160 bp and 31 bp and AG genotypes 160 bp, 31 bp and 91 bp. The results indicate that appropriate sequencing or RFLP typing can be performed.

### 3.3. RFLP Detection

The digestion G1-RFLP and BMPR1B-RFLP amplification products were electrophoresed, and the electrophoresis results are shown in Figure 1 and Figure 2. The results showed that G1 and FecB mutations were not found in any of the 216 samples.

**Figure 1 genes-14-01102-f001:**
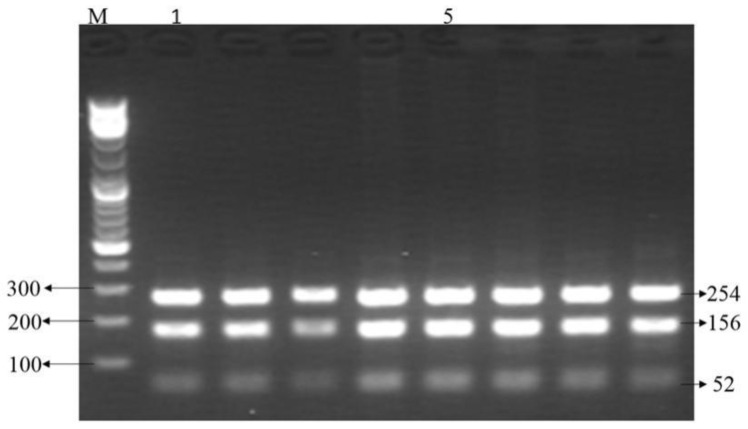
Digestion results of G1-RFLP amplification product, M: Marker (BBI 1 KB Plus (0.1–10 kb) DNA Ladder). The electrophoresis band is in bp.

**Figure 2 genes-14-01102-f002:**
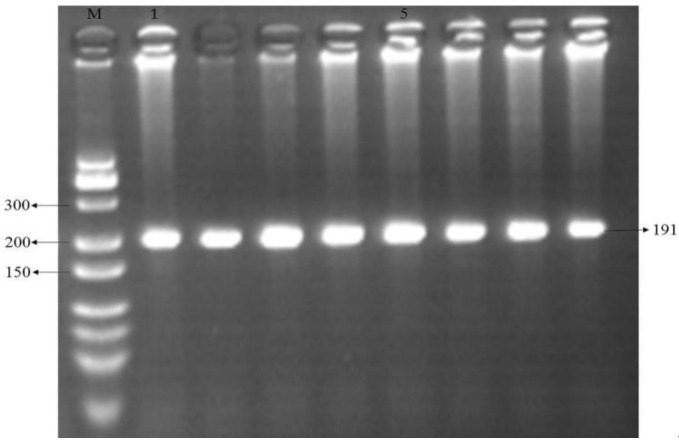
PCR-RFLP with primer BMPR1B-RFLP (3% agarose gel), M: Marker (BBI BBI Low MW DNA Marker-A). The electrophoretic band unit is bp.

### 3.4. Amplified Fragment Sequencing

There were two overlapping peaks in the amplification fragment peak patterns of GDF9-1 and GDF9-2 primers (Appendix A), located at C→T at coding sequence 719 bp and C→A at 1189 bp, respectively. In which C→T mutations at 719 bp and C→A mutations at 1189 bp lead to changes in the amino acids encoded by the corresponding sites (Table 3), and the polymorphic sites G2, G3, G4, G5, G6, G7, G8, etc., found in sheep covered by the amplification bands FecG^E^ and FecTT sites were checked, and it was found that all samples had G3 and G4 homozygous mutations.

Sequencing analysis of GDF9-1, GDF9-2 and BMP15-1 amplification products found that there were overlapping peaks in the sequence peak map of the BMP15-1 amplification products (Appendix A), so it could be determined that there were SNP sites, which were located at G→A at 732 bp and C→G at 805 bp, where C→G mutations at 805 bp led to changes in the amino acids encoded by the corresponding sites (Appendix A). At the same time, the polymorphism sites B2, B3, B4, FecX^H^, FecX^I^ and FecX^L^ found in sheep covered by the amplification band were checked and no mutations were found.

### 3.5. Polymorphism Statistics

The statistics of the four polymorphisms detected and the allele frequency, genotype frequency, homozygosity and heterozygosity were calculated, and the statistical results showed that the G732A loci of the BMP15-1 amplification fragment of 216 samples were dominated by the GG genotype, 150 of the 216 individuals were GG genotypes and only 5 individuals were homozygous mutants (see Table 4). The G allele frequency was 0.836, and the C805G locus was dominated by the CC genotype; the C719T locus of the *GDF9* amplification fragment of 216 samples was dominated by the CC genotype, with a genotype frequency of 0.944, only 12 individuals were CT genotypes and no homozygous mutant TT genotype was found, while the G1189A locus was dominated by the GG genotype.

## 4. Discussion

In small ruminant species, mainly in sheep, the genetics of litter size is extensively studied. In sheep, the main prolificacy genes include *BMPR1B*, *GDF9* and *BMP15*. The mutated *BMPR1B* gene increases the ovulation rate. Litter size and ovulation rate increase with the number of copies of the mutation. In heterozygotes, there are high ovulation rates in *GDF9* and *BMP15* mutants while the homozygous mutants show a primary ovarian failure, causing complete sterility [31]. The Tibetan cashmere goat with high prolificacy varies from twin to quadruplet offspring. In this study, the goats were genotyped for three known prolificacy genes, namely, *BMPR1B*, *GDF9* and *BMP15*. SNP genotyping techniques rely on amplification of the target DNA using PCR but different by discriminating between the different alleles, and this includes significant post-PCR manipulation. The restriction fragment length polymorphism (RFLP) typing method involves restriction endonuclease digestion of PCR products. A well-used SNP typing technique, Allele-specific oligonucleotide (ASO) melting, is characterized by lengthy blotting and hybridization procedures. The use of the tetra-primer ARMS-PCR method used circumvents much handling of the PCR product. Primers were designed to amplify fragments of differing sizes for each allele band for the purpose of resolving them in agarose gel electrophoresis. It is a simple, fast and highly economical method for SNP scoring and very important tool for large-scale SNP analysis [31].

The current study, examining the G1 mutation in the *GDF9* gene, mainly focuses on sheep, whereas if the G1 mutation in the *GDF9* gene of different breeds of sheep affects litter size in different breeds of sheep is currently different. G1 mutation in different breeds of sheep and goats was a little different, and its effects on litter size attracted more interest [8,32,33,34]. The RFLP typing results of the G1 mutation in the *GDF9* gene of Tibetan cashmere goats show that no mutation exists at this site, which is similar to other studies [35] Consistency in the presence of mutations at this site was also not found in cashmere goats in Inner Mongolia.

Polley et al. [35] found that FecB mutations had a vital effect on litter size in Indian black Bengal goats in *BMPR1B*. However, in related studies of other breeds of goats, most of them did not have this locus mutation, and this study also showed that there were no *FecB* mutations in Tibetan cashmere goats, indicating that the mutation rate of this site was low in goats.

The detection of known polymorphisms in sheep covered in the amplification fragments of Tibetan cashmere goats GDF9-1, GDF9-2 and BMP15-1 only found the presence of G3 and G4 mutations, and both were homozygous mutations. It was inferred from the mutation rate of G3 and G4 that their mutations did not affect reproductive performance in Tibetan cashmere goats. G3 and G4 mutations have not been found to affect litter size in some breeds of sheep and goats because G3 does not cause amino acid changes, and G4 mutations located before the Flynn protease cleavage site do not affect protein function.

In recent years, *BMP15*, *GDF9*, *BMPR1B*, insulin-like growth factor 1 (*IGF1*), follicle-stimulating hormone β (*FSHβ*) and prolactin receptor (*PRLR*) have been the main genes affecting fertility, as demonstrated in sheep and other species [36,37,38]. According to the results of the current study, these are known mutations (B2, B3, B4, FecX^H^, FecX^I,^ FecX^L^, G1, G2, G5, G6, G7, G8, FecG^E^, FecTT and FecB, etc.) affecting litter size in goats. These mutations are rarely found in both high-yielding goat breeds and low-yielding goat breeds, and this may also indicate that polymorphic sites affecting the shape of litter size in sheep and goats may be different [37,39,40,41].

In the detection of the polymorphism of the amplification fragment of BMP15-1 in Tibetan cashmere goats, two new mutation sites, G732A and C805G, were found, in which the G732A mutation did not lead to changes in amino acids and had no effect on its protein structure. Therefore, it is possible that G732A mutation may not have affected reproductive performance [42,43]. The C→G mutation at the C805G locus caused the amino acid encoded by the corresponding site to change from hydrophilic glutamine to hydrophilic glutamate, and from polar uncharged amino acid to negatively charged amino acid, which may have a certain impact on the secondary structure of its protein. Relevant studies have shown that Tibetan cashmere goats are generally singleton [44], and only a very small number of individuals have twins. From the results of G732A and C805G, two sites of homozygous mutations [45] are relatively small, 5 and 13, and it is impossible to judge whether their homozygous mutations will have an impact on the litter size, respectively. From the heterozygous mutations [46] of these two sites, it can be inferred that the heterozygous mutations may not have much of an effect on litter size [47].

The detection of GDF9-1 and GDF9-2 amplified fragment polymorphisms found that there were two polymorphic sites, C719T and G1189A, in Tibetan cashmere goats. Existing studies have shown that C719T and G1189A mutations were found in goat breeds, such as Guizhou white goat, Indian goat, Jining green goat and Boer goat. Du Zhiyong et al. [31], in a study of Guizhou white goats, believed that the G1189A mutation might be related to its high fertility. Dong Chuanhe et al. [48] found that G1189A mutation had a significant effect on the litter size in Yimeng black goats, the litter size of the AA type was 0.34 and 0.38 more than that of the AG and GG types, respectively, and C719T had no significant effect on the litter size. In Tibetan cashmere goats, only heterozygous mutations existed with a low frequency of 0.056. It was also uncertain whether it could affect the litter size [49] of Tibetan cashmere goats or not. The frequency of G1189A homozygous mutant type was 0.116, and it was speculated that the homozygous mutation at this site might affect the litter size of Tibetan cashmere goats. 

## 5. Conclusions

This study reveals that there were no G1, B2, B3, B4, FecX^H^, FecX^I^, FecX^L^, G2, G5, G6, G7, G8, FecG^E^, FecTT and FecB mutations found in Tibetan cashmere goats; however, G3 and G4 mutations were detected in the *GDF9* gene and were all homozygous mutations in the 216 samples, which were significantly different from those in sheep. A total of four polymorphisms, C719T, G1189A, G732A and G1189A, were found in the *BMP15* and *GDF9* genes of Tibetan cashmere goats, and the two polymorphism sites of G732A and G1189A in the *BMP15* gene were discovered. These results enrich the mutation data of *BMP15*, *GDF9* and *BMPR1B* in goats and also provide data for future research on high-fertility gene biomarkers of goats.

## Figures and Tables

**Table 1 genes-14-01102-t001:** The primer information.

Primer Name	Primer Sequence (5′→3′)	Annealing Temperature (°C)	Product Size (bp)
GDF9-1 GDF9-2 G1-RFLP BMP15-1 BMPR1B-RFLP	F-GAACCTTTCCATCAGTG R-TCAGATACAAAAGCAGTG F-TATCTGCCTACCCCGTGG R-GGTACACTTAGTGGCTATCA F-GAAGACTGGTATGGGGAAATG R-CCAATCTGCTCCTACACACCT F-AGCAGCCAAGAGGTAGTGAGG R-CATGTGCAGGACTGGGCAATC F-CCAGAGGACAATAGCAAAGCAAA R-CAAGATGTTTTCATGCCTCATCAACACGGTC	53 56.5 60 61.3 60	408 461 466 550 191

**Table 2 genes-14-01102-t002:** PCR amplification system.

Reagents	Volume (μL)
10× buffer	2.5
Mg^2+^ (10 mol/L)	2.0
dNTP (2 mmol/L)	2.0
Primers	1
TaqDNA polymerase (5 U/μL)	0.13
DNA template	1
ddH_2_O	Make up 25

**Table 3 genes-14-01102-t003:** Positions of mutations with changes in bases and the effect with amino acids in *GDF9* and *BMP15* of Tibetan cashmere goat.

Gene	Mutation	Position (bp)	Base Change	Codon	Amino Acid Change
*GDF9*	G3	477	G—A	CTG*—CTA	Leucine (L) unchanged
	G4	721	G—A	G*AA—AAA	Glutamic acid (E)—Lysine (K)
	C719T	719	C—T	GC*G—GTG	Alanine (A)—Valine (V)
	G1189A	1189	G—A	C*TT—ATT	Valine (V)—Isoleucine (I)
*BMP15*	G732A	732	G—A	AAG*—AAA	Lysine (K) unchanged
	C805G	805	C—G	C*AA—GAA	Glutamine (Q)—Glutamic acid (E)

* Locus of changed nucleotide.

**Table 4 genes-14-01102-t004:** Genotype and allele frequencies of *GDF9* and *BMP15* gene in Tibetan cashmere goats.

Gene	Mutation	Genotypes	Frequencies	Alleles	Frequencies	Heterozygosity	Homozygosity
*GDF9*	G3	AA (N = 216)	1.000	A	1.000	0.000	1.000
	G4	AA (N = 216)	1.000	A	1.000	0.000	1.000
	C719T	CC (N = 204)	0.944	C	0.972	0.056	0.944
		CT (N = 12)	0.056	T	0.028		
	G1189A	GG (N = 125)	0.579	G	0.731		
		GA (N = 66)	0.305			0.306	0.694
		AA (N = 25)	0.116	A	0.269		
*BMP15*	G732A	GG (N = 150)	0.695	G	0.836		
		GA (N = 61)	0.282			0.282	0.718
		AA (N = 5)	0.023	A	0.164		
	C805G	CC (N = 134)	0.620	C	0.780		
		CG (N = 69)	0.320			0.319	0.681
		GG (N = 13)	0.060	G	0.220		

## Data Availability

The data presented in this study are available on request from the corresponding author.

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
