# Peer review of "Polymorphisms Analysis of BMP15, GDF9 and BMPR1B in Tibetan Cashmere Goat (Capra hircus)"

_genes, 2023, doi:10.3390/genes14051102_

Round 1
Reviewer 1 Report
Revision of manuscript entitled “Polymorphisms analysis of BMP15, GDF9 and BMPR1B in Tibetan cashmere goat (Capra hircus)”.
The topic is of interest to the field, but the data reported in the manuscript is too limited.
The Abstract section is confusing, punctuation is missing, and some elements are not clear. Please clearly state that the genes you analyzed are candidate genes in sheep. Please clarify which SNPs were genotyped: were they already known in the goat species? Weren't they known, and you found them for the first time? If so, please explain how you discovered them.
The Introduction section is confused as well, the authors explain all known variants in sheep (long list), but they don't explain the state of the art in goat. Are there any studies on prolificacy genes in goat? These should be stated in the introduction section, at least in a concise way.
Lines 58-62: Please explain what your research hypothesis is and what your research goal is, at the end of the introduction section.
Materials and methods section: do you have the approval of an ethics committee from your research institution?
How prolific were the goats you analyzed? What parity were they in? What was their diet and their level of health and well-being? How were they farmed (intensive or extensive)?
Line 83: why did you choose exon II? Did you do an alignment between goat and sheep sequences before starting genotyping? Why did you build the primers on the sheep sequence and not on the goat sequence?
Line 98-101: please do not repeat the information which is already given in Tables.
Figure 3 and Figure 4. These two Figures might be reported as supplementary material.
Table 3. Polymorphisms need to be reported in a different way, the authors should check to see if their mutations already have an #rs SNP ID.
In my opinion, the english language is acceptable.
Author Response
Dear Editor and Reviewer,
Thank you for the suggestions and comments. They have been closely followed and revisions have been made accordingly. The following are the reviewer’s comments along with our summarized responses.
Reviewer 1:
Comments and Suggestions for Authors
Revision of manuscript entitled “Polymorphisms analysis of BMP15, GDF9 and BMPR1B in Tibetan cashmere goat (Capra hircus)”. The topic is of interest to the field, but the data reported in the manuscript is too limited.
- The Abstract section is confusing, punctuation is missing, and some elements are not clear. Please clearly state that the genes you analyzed are candidate genes in sheep. Please clarify which SNPs were genotyped: were they already known in the goat species? Weren't they known, and you found them for the first time? If so, please explain how you discovered them.
Response: This has been corrected and the sentences have been improved and made clear. The SNP genotype were found in goat species.
- The Introduction section is confused as well, the authors explain all known variants in sheep (long list), but they don't explain the state of the art in goat. Are there any studies on prolificacy genes in goat? These should be stated in the introduction section, at least in a concise way.
Response: This section has been checked and improved. Studies in prolificacy in goats have been included.
The mutation of BMP15 genes associated with the fecundity in goats have been confirmed in Markhoz goats [1].
There is a high influence of BMP15 and GDF9 on fecundity. They induce mitosis along with differentiation in the follicular somatic cells during follicular development through paracrine signaling pathway. They also play key roles in litter size. Numerous mutations in these genes may contribute to high prolificacy [1] [2].
FecXB has been found in goats. It has also been shown that the genetic polymorphisms of the GDF9 gene in goats were significantly higher than in sheep and also more complicated [3].
FecB mutation in BMPR1B gene has also been found in many goat breeds. A previous study found them to be homozygous non-carriers. FecB polymorphism in Raighar goat showed monomorphic status of the breed. In Markhoz goat, it was supposed that FecB allele might be not the only main cause for prolificacy rate in goat [2].
- Lines 58-62: Please explain what your research hypothesis is and what your research goal is, at the end of the introduction section.
Response: This study focuses on the effects of polymorphism and mutation of genes (BMP15, GDF9, and BMPR1B) on fertility in Tibetan cashmere goat. Identifying genes associated with fertility of these goats is important for their breeding, better fertility and normal physiology.
- Materials and methods section: do you have the approval of an ethics committee from your research institution?
Response: All experimental procedure involving animals were reviewed and approved by the Animal experimental committee of Sichuan Agricultural University. The ethical approval code is (SAU2015068, Dec 12, 2015).
- 5. How prolific were the goats you analyzed? What parity were they in? What was their diet and their level of health and well-being? How were they farmed (intensive or extensive)?
Response: There were single birth animals with three to five consecutive parities. All the goats were healthy and aged 3-4 years. They were kept in sheltered outdoor paddock and were provided with alfalfa hay and water available ad libitum.
- Line 83: why did you choose exon II? Did you do an alignment between goat and sheep sequences before starting genotyping? Why did you build the primers on the sheep sequence and not on the goat sequence?
Response: BMP15 genotype were determined by analysis of three nucleotide fragments that spanned most of exon 2. GDF9 genotypes were determined by analysis of five nucleotide fragments which spanned exon 1, part of the intron and most of exon 2. We did an alignment between goat and sheep sequence and found out that the similarities of the two genes were very high. The sequences of primers we used for the genes between goat and sheep were the same.
“According to the BMP15 gene sequence of sheep published in GenBank (accession number: EU743938), the primer premier 5.0 software was used to specifically amplify primers of BMP15-1 for the exon II segment, and its amplification region covered the known B2, B3, B4, FecXH, FecXI and FecXL alleles in sheep. Specific primers of GDF9-1 and GDF9-2 were designed for exon fractions based on published sheep GDF9 gene sequences (accession number: EF446168) using primer premier 5.0 software and their amplification regions covered known G2, G3, G4, G5, G6, G7, G8, FecGE and FecTT alleles in sheep.”
- Line 98-101: please do not repeat the information which is already given in Tables.
Response: This has been corrected and the repeated information has been deleted.
- Figure 3 and Figure 4. These two Figures might be reported as supplementary material.
Response: These two figures have been reported as supplementary materials.
- Table 3. Polymorphisms need to be reported in a different way, the authors should check to see if their mutations already have an #rs SNP ID.
Response: Thank you for the comment. This would be considered in our next paper.
Reviewer 2 Report
line 71-72 - if in case of other manufacturers/products you descibe full name of them here please add also full name of profuct and manufacturer with country of origin
line 100-102 - I have serious doubts if this PCR profile will work especially for primers amplifying product with over 400 bp - normally manufactures suggest (and also from my lab experience) that over 400 bp extension time should be at least 1 minute
109 -110 - restriction enzymes were manufactred by?? and protocol of digestion? also those enzymed were from some publication or it was you who design this PCR-RFLP?
Figure 1 and 2 - why do you used different DNA marker?
table 4 - Honozygosity? i assume there should be homozygosity/ please add there some spacing to make clearly which genotpyes are for which mutation, it's little unclear e.g. for G4 mutation it seems that CC(N-204) genotypes is included
Discussion:
Why authors did no collected any informations about fertility traits from Cashmere goats/number of kids? in this form this manuscript is less likely intresting as it is only a strict analysis of sequencing data plus genotyping. If you did not have any data for association anaylis at least some bioinformatics analysis should be included.
Author Response
- line 71-72 - if in case of other manufacturers/products you describe full name of them here please add also full name of product and manufacturer with country of origin.
Response: “….. from Beijing Tiangen Biotechnology, China. Molecular reagents, such as Taq DNA polymerase and dNTP were purchased from Dalian TAKARA Corporation (Dalian, Liaoning, China) and used according to the manufacturer’s instructions.”
- line 100-102 - I have serious doubts if this PCR profile will work especially for primers amplifying product with over 400 bp - normally manufactures suggest (and also from my lab experience) that over 400 bp extension time should be at least 1 minute
Response: According to the manufacturer’s protocol, 1 kb will use 1min and for 500 bp, 30s is enough.
- 109 -110 - restriction enzymes were manufactred by?? and protocol of digestion? also those enzymed were from some publication or it was you who design this PCR-RFLP?
Response: The restriction enzymes were manufactured by Dalian TAKARA Corporation (Dalian, Liaoning, China). Protocol of digestion, The products amplified by G1-RFLP and BMPRIB-RFLP were digested with Hha I and Ava enzyme seperately and the total volume was adjusted using products 10 μL, 10×Buffer R 1.5 μL, enzyme (10U/μL) 0.1 μL, ddH2O 3.4 μL, 37°C overnight. The ezymes used were from a previous publication [4].
- Figure 1 and 2 - why do you used different DNA marker?
Response: We used different markers, however both DNA markers meet the requirements of our experiment. The first marker finished after using it for Figure 1 hence, we used a second (different) marker for the Figure 2.
- table 4 - Honozygosity? i assume there should be homozygosity/ please add there some spacing to make clearly which genotpyes are for which mutation, it's little unclear e.g. for G4 mutation it seems that CC(N-204) genotypes is included.
Response: Revised and improved. Honozygosity has been changed and corrected to “Homozygosity”.
From the Table 4, we can see the total number was N=216, So AA(N=216) genotypes is included only in G4 mutation and CC(N=204) and CT(N=12) genotypes is included in C719T mutation.
Discussion:
Why authors did no collected any informations about fertility traits from Cashmere goats/number of kids? in this form this manuscript is less likely interesting as it is only a strict analysis of sequencing data plus genotyping. If you did not have any data for association analysis at least some bioinformatics analysis should be included.
Response: The results and discussion is to enrich the mutation data of BMP15, GDF9 and BMPR1B in goats, and also provide data for future research on high fertility gene biomarkers of goats. We would consider to write more about fertility traits in our future works.

Round 2
Reviewer 1 Report
The authors responded to the indications and improved the manuscript.
Author Response
Responded.
Reviewer 2 Report
In lines 85 - 88 i believe that there is aim of study - I did not find any results proving this? where are results of effect on fertility?
Why from GDF9 only second exon was sequenced and without UTR part? You wrote that you want to enrich knowledge about polymorhisms and yet you did not sequenced first exon?
for BMP15 - only part of exon 2 was sequenced - what with rest of gene?
Also how did you idenified polymophisms? what analysis did you perfrom?what tools you used to check quality of reads and how do you performed aligment?
for BMP15 - you stated that for primers design you used EU743938 record - according this record CDS is between 144 and 471 bp and 5781 and 6637 bp - how than positions 732 and 805 can code aminoacids if they are within intron? even checking ensembl database and BMP15 record on 732 position there is G nucleoide but there is L (leucine) aminoacid??!!
Table 2 is PCR amplification mixture and why in line 190 you stated that there are some informations about SNPs?
lines 252-254 - The C→G mutation at the C805G locus caused the amino acid encoded by 252 the corresponding site to change from hydrophilic glutamine to hydrophilic glutamate, 253 and from polar uncharged amino acid to negatively charged amino acid, which may have 254 a certain impact on the secondary structure of its protein. - and this statement is based on what? there are plenty of bioinformatic tools (Project HOPE, PANTHER, PolyPhen2 - it is for human but works on other species) to eveluate such a influence whch should be done here. It seems that whole analysis is performed based on "which may have infleuce" statement which is not acceptable
line 266 - believed that the mutation has impact of performed association study or/and some bioinformatic analysis?
Author Response
Dear Editor and Reviewer,
We have significantly improved and clarified certain points in manuscript with respect to the main concerns of the reviewer. The comments have been closely followed and revisions and clarifications have been made accordingly. The following are the reviewer’s comments along with our responses.
Reviewer’s comments
- In lines 85 - 88 i believe that there is aim of study - I did not find any results proving this? where are results of effect on fertility?
Response: This study focuses on the effects of polymorphism and mutation of genes (BMP15, GDF9 and BMPR1B) in Tibetan cashmere goat. It is advantageous to identify genes and mutations that can be important for their breeding, better fecundity and normal physiology to improve productivity and reproductive traits.
- Why from GDF9 only second exon was sequenced and without UTR part? You wrote that you want to enrich knowledge about polymorhisms and yet you did not sequenced first exon?
Response: We focused on polymorphism of GDF9. There was no polymorphism in UTR region so the first exon (G1-RFLP primers) and second exons were sequenced. We amplified and sequenced G1 from the first exon but there was no polymorphism.
- for BMP15 - only part of exon 2 was sequenced - what with rest of gene?
Response: From previous publications, the known B2,B3, B4, FecXH, FecXI and FecXL alleles in sheep were from exon 2 and they are identified as fecundity related mutations. Therefore we amplified exon 2 in order to detect the polymorphisms of BMP15 gene. We amplified exon 1, 2 and intron of BMP15 gene and the NCBI accession number was KY780297.1.
- Also how did you idenified polymophisms? what analysis did you perfrom? what tools you used to check quality of reads and how do you performed aligment?
Response: There were two ways to identify polymorphism in this study. (1) Primer amplification products of GDF9-1, GDF9-2 and BMP15-1 were sequenced after electrophoresis test. The sequencing GDF9 and BMP15 was obtained from the company. According to the sequencing from the peak figure (supplementary figure), overlapping peaks detected by two kinds of nucleotide signals in a certain base location were noted as SNP site. (2) The products amplified by G1-RFLP and BMPR1B-RFLP were digested with Hha I and Ava II enzymes. Hha I enzyme cleavage G1-RFLP product typing bands were GG genotypes (wild type) 254 bp, 156 bp and 52 bp, AA genotypes 414 bp and 52 bp, GA genotypes 414 bp, 254 bp, 156 bp and 52 bp. The Ava II enzyme cleavage BMPR1B-RFLP product typing bands were AA genotypes (wild type) 191 bp, GG genotypes 160 bp and 31 bp, and AG genotypes 160 bp, 31 bp, and 91 bp. The results indicated that appropriate sequencing or RFLP typing could be performed.
- for BMP15 - you stated that for primers design you used EU743938 record - according this record CDS is between 144 and 471 bp and 5781 and 6637 bp - how than positions 732 and 805 can code aminoacids if they are within intron? even checking ensembl database and BMP15 record on 732 position there is G nucleoide but there is L (leucine) aminoacid??!!
Response: We took out exon 1 and 2 and put them together without intron and reordered the sequence of BMP15 to perform the CDS sequence analysis. When the G nucleotide in 732 position became A, the coding amino acid lysine (K) didn’t change (AAG-AAA).
- Table 2 is PCR amplification mixture and why in line 190 you stated that there are some informations about SNPs?
Response: We have added a supplementary table to make this point clear.
Supplementary Table 1.
|
Mutation |
Position(bp) |
Base change |
Codon |
Amino acid change |
|
G732A |
732 |
G-A |
AAG*-AAA |
Lysine (K) unchanged |
|
C805G |
805 |
C-G |
C*AA-GAA |
Glutamine (Q) - Glutamic acid (E) |
- lines 252-254 - The C→G mutation at the C805G locus caused the amino acid encoded by 252 the corresponding site to change from hydrophilic glutamine to hydrophilic glutamate, 253 and from polar uncharged amino acid to negatively charged amino acid, which may have 254 a certain impact on the secondary structure of its protein. - and this statement is based on what? there are plenty of bioinformatic tools (Project HOPE, PANTHER, PolyPhen2 - it is for human but works on other species) to eveluate such a influence whch should be done here. It seems that whole analysis is performed based on "which may have infleuce" statement which is not acceptable
Response: The BMP15 gene had two new mutation sites G732A and C805G. The G732A mutation did not lead to change in amino acids and had no effect on its protein structure. With the C805G site, we can only make speculations about the changes from Glutamine (Q)- Glutamic acid (E) with the hydrophilic- hydrophilic, polar uncharged-negatively charged and their influence on litter size.
- line 266 - believed that the mutation has impact of performed association study or/and some bioinformatic analysis?
Response: This line has been deleted for clarity.